

# Impact of topography on in-situ soil wetness measurements for regional landslide early warning – a case study from the Swiss Alpine Foreland

Adrian Wicki[1], Peter Lehmann[2], Christian Hauck[3], Manfred Stähli[1]

[1]Swiss Federal Institute for Forest, Snow and Landscape Research WSL, Zürcherstrasse 111, 8903 Birmensdorf, Switzerland
[2]ETH Zurich, Institute of Terrestrial Ecosystems, Universitätstrasse 16, 8092 Zürich, Switzerland
[3]University of Fribourg, Department of Geosciences, Chemin du Musée 4, 1700 Fribourg, Switzerland

*Correspondence to*: Adrian Wicki (adrian.wicki@wsl.ch)

**Abstract.** Recent studies have demonstrated the potential of in-situ soil wetness measurements to predict regional shallow landslides. Increasing availability of monitoring data from sensor networks provides valuable information for developing future regional landslide early warning systems (LEWSs), however, most existing monitoring sites are located on flat terrain. The question arises, if the representativeness for regional landslide activity would improve if sensors were installed on a landslide-prone hillslope? To address this, two soil wetness monitoring stations were installed at close proximity on a steep slope and on a flat location in the Napf region (Northern Alpine Foreland of Switzerland), and measurements were conducted over a period of 3 years. As both sites inhibit similar lithological, vegetation and precipitation characteristics, soil hydrological differences can be attributed to the impact of topography and hydrogeology. At the sloped site, conditions were generally wetter and less variable in time, and evidence was found for temporary lateral water transport along the slope. These differences were systematic and could be reduced by considering relative soil moisture changes. The application of a statistical landslide forecast model showed that both sites were equally able to distinguish critical from non-critical conditions for landslide triggering, which demonstrates the value of existing monitoring sites in flat areas for the application in LEWSs.

## 1 Introduction

Landslides are a frequent natural hazard in mountainous regions all around the world, causing fatalities and damages to infrastructure and buildings every year. (Froude and Petley, 2018) The term 'landslide' refers to various types of mass movements including different source materials, process dynamics and triggering factors (Hungr et al., 2014; Varnes, 1978); here we focus on shallow landslides triggered by rainfall or snow-melt water infiltration. Shallow landslides are mostly triggered by the direct infiltration from the surface, lateral movement or exfiltration of water from the bedrock to the subsurface, and the consequent rise or formation of (perched) water tables. This results in a short-term decrease in suction or increase in pore water pressure, and the eventual decrease in shear strength below a critical value for failure. (Van Asch et al., 1999; Godt et al., 2009; Terzaghi, 1943) The disposition of a slope to fail is mainly determined by lithological characteristics



of the source material (mechanical properties and material thickness), the slope angle, hydrogeological characteristics of the
      slope (i.e. the ability of a slope to take up, route and drain water), and the nature of the vegetation cover. These factors are
      controlled by processes that act over long time scales and thus are often used in susceptibility analysis. (Brabb, 1985) On
      shorter time scales, the seasonal wetting and drying up of a slope control the variable disposition to failure, whereas the
      eventual triggering is related to the infiltration of rainfall or snow melt water and happens on short scales of minutes to days.

(Bogaard and Greco, 2016)
      Regional landslide early warning systems (LEWSs) are used to assess the temporally varying landslide danger in a defined
      territorial unit. (Guzzetti et al., 2020; Piciullo et al., 2018) In recent years and decades, they have become an essential and
      reliable instrument for authorities to issue alerts, and thus permit to move people or mobile goods at risk to safety. (Stähli et
      al., 2015) LEWSs use empirical rules relating the temporal variation of hydro-meteorological variables such as rainfall, soil

wetness or groundwater levels to the regional occurrence of landslides and are thereby able to discriminate critical
      environmental conditions for landslide triggering. In the past, most regional LEWSs have been based on widely available
      rainfall data and the description of precipitation event characteristics such as intensity, duration, or total amounts. (e.g. Caine,
      1980; Guzzetti et al., 2008) However, it has long been recognised that in places with a strong seasonal soil wetness cycle the
      predisposition of slopes to fail may change significantly (Thomas et al., 2020), e.g. by reducing critical precipitation amounts

needed for landslide triggering (Ashland, 2021; Baum and Godt, 2010) or by altering the hydrological connectivity to the
      draining bedrock (Greco et al., 2021). In regions with such seasonal variations, information on rainfall is not sufficient to
      discriminate critical conditions and defining empirical rules of LEWSs. But with the growing availability of soil wetness
      information, hydro-meteorological LEWSs have emerged that often combine precipitation thresholds with soil wetness
      information. (Bordoni et al., 2021; e.g. Mirus et al., 2018b; Zhao et al., 2019)

In this context, the wetness of the soil is commonly described by two variables. The volumetric water content (VWC), $\theta$ ($m^3$
      $m^{-3}$), describes the water content on a volume basis and is defined as ratio of the volume of water, $V_w$, to the bulk volume of
      soil, $V_b$. The soil water potential (SWP), $\psi$ (Pa), describes the energy state of soil water compared to the energy state of free
      water at reference level. The soil water potential includes the matric potential, $\psi_m$ (hPa), in the unsaturated soil (and in the
      saturated region of the capillary fringe), and the pressure potential, $\psi_p$ (hPa), that characterizes saturated conditions below a

water table (note that the osmotic potential is neglected in this study). The matric potential results from capillarity and
      adsorption of the soil matrix and ranges from large negative SWP values to $\psi = 0$. The pressure potential describes the
      hydrostatic pressure of an overlying water table and ranges from $\psi = 0$ to positive values.
      Both VWC and SWP can be measured in-situ at the scale of single vertical profiles (often referred to as "point scale"). VWC
      is usually estimated by electromagnetic sensors that generate an electric field along parallel electrodes or pairs of rings. The

resulting electromagnetic properties, such as the travel time of a step voltage pulse (in case of time domain reflectometry TDR)
      or the charge time (in case of capacitance-based sensors), are then related to the dielectric permittivity of the bulk soil.
      (Babaeian et al., 2019) In return, VWC is typically estimated from dielectric permittivity values by empirical calibration
      functions that make use of the strong contrast of electrical permittivity of water (80), minerals (3-5) and air (1). (e.g. Topp et





al., 1980) Data quality depends on the measurement frequency, soil temperature and clay content of the soil, and precision can
be improved by applying site-specific calibration functions. (Kelleners et al., 2005; Robinson et al., 2008) SWP is commonly
measured by tensiometers, which consist of a porous cup arranged at the end of a water-filled rigid tube. If the porous cup is
in contact with the soil, water may escape or enter the tensiometer until it is in equilibrium with the soil water. The resulting
water pressure is measured by a vacuum gauge or pressure transducer. (Livingston, 1908; Or, 2001) Under (unsaturated)
conditions with $\psi_m$<0, water flows out of the tensiometer and creates suction whereas under saturated conditions ($\psi_p$>0),
positive value readings result. Measurement of SWP is limited by the vaporisation point of water at approximately -1000 hPa,
thus the application of tensiometers is limited to medium to very wet conditions. (van der Ploeg et al., 2010) In-situ soil wetness
sensors may be affected by local heterogeneities, such as the presence of macropores or lithological differences (Beven and
Germann, 2013) due to the relatively small measurement volumes of up to a few litres of soil (Jackisch et al., 2020). To increase
the robustness and representativeness of the measured signal, in-situ sensors are often combined to sensor arrays or sensor
networks across various depths or locations. (Robinson et al., 2008; Vereecken et al., 2014)

Topography heavily influences the spatial distribution of soil moisture by controlling the partitioning of infiltration and surface
runoff (Cerdà, 1997; Fox et al., 1997), as well as the lateral movement of soil water and groundwater along the slope direction,
and accumulation and drainage of water in specific zones along the hillslope (Freer et al., 2002; e.g. Sidle et al., 2000). Lateral
water flow within a hillslope often occurs within hydrologically connected saturated or nearly saturated layers and areas. Such
saturation layers can form if a highly permeable layer is situated above an impeding layer such as the bedrock or an argillic
horizon (e.g. Tani, 1997). Alternatively, lateral preferential flow may occur along slope-parallel macropores or high-permeable
layers with coarse texture and large pore space (Beven and Germann, 1982; Tsuboyama et al., 1994), or by water exchange
with the underlying fractured bedrock (Brönnimann et al., 2013; Montgomery et al., 1997). Slope and aspect further control
rates of evapotranspiration and snow melt by solar radiation differences. (McVicar et al., 2007; e.g. Xu et al., 2004) Thus, the
placement of a soil wetness monitoring site in a complex landscape is expected to have an impact on the measured values of
VWC and SWP, but potentially also on storm-event properties that are derived for warning purpose.

In hydrology, the topographic influence on soil water distribution (e.g. groundwater levels or soil moisture variation) has often
been modelled by the topographic wetness index (TWI). The TWI combines the amount of water flowing to a specific point
(approximated by the upslope contributing area) and the subsurface lateral transmissivity (approximated by the local slope).
(Beven and Kirkby, 1979) However, the application of the TWI is limited by the steady state formulation and the results
strongly depend on the selected TWI approach and parametrisation (Kopecký et al., 2021; Sørensen et al., 2006). Alternatively,
soil water distribution can be obtained by the application of distributed hydrological models which allow dynamic simulations
of soil hydrological processes, for which additional input parameters are required. (Grabs et al., 2009) These approaches are
based on the assumption that surface topography is the main controlling factor for groundwater table depths and typically
neglect other influencing factors such as bedrock topography (Freer et al., 2002).

The complexity of the flow patterns and the local variations of climatic conditions challenge the definition of the relationship
between landslide activity and critical hillslope wetness patterns that could be implemented in LEWSs. In a previous study,





we related the temporal variation of in-situ soil moisture to the regional landslide activity and we found that the correlation increased with decreasing distance from the soil moisture site. (Wicki et al., 2020) Representativeness of in-situ measurements for critical conditions was increased by normalizing VWC values to the degree of saturation and by integrating sensors across various soil depths. The monitoring sites collected for the study were mostly located on flat or moderately sloped terrain, because they were installed with different purposes than landslide early warning, such as improving meteorological forecasts, drought monitoring, soil conservation, or permafrost monitoring. (Stehrenberger and Huguenin-Landl, 2016) Hence, they are potentially affected by different soil hydrological processes than such that occur on landslide-prone hillslopes. The question arises, if hydrological-based LEWSs would be more reliable if soil wetness was measured on slope locations. Or in other words, could the representativeness for regional landslide activity be increased if soil moisture was measured at places that are susceptible to the landslide process?

To address these questions, two soil wetness monitoring stations were installed at a hillslope site prone for shallow landslides and at a nearby flat location. Both sites comprise similar lithological and vegetational characteristics and they receive roughly the same precipitation amounts. Differences in soil hydrological conditions can thus be mainly attributed to the effect of topography. Here, we focus on the quantification of differences in measured soil wetness variation at the two sites. Further, by application of an empirical landslide forecast model, the ability to predict regional landslide activity is compared. This work aims at providing a basis to decision makers on designing future soil wetness monitoring systems in regional LEWSs.

## 2 Material and methods

### 2.1 Study site

The study site is located near the village of Wasen in the Napf region (Switzerland, Figure 1), a mountainous area at the northern edge of the Alps. The elevation ranges from 600 to up to 1404 m a.s.l. (Napf peak), and the landscape is characterized by steep terrain, which was mainly formed by fluvial erosion since the Pleistocene era. (Schlüchter et al., 2019) Climatological mean annual precipitation amounts for the period 1991–2020 are 1400 and 1600 mm at the nearby meteorological stations Wasen (2.8 km distance) and Kurzeneialp (2.0 km distance), respectively. (MeteoSwiss, 2022) Precipitation falls throughout the year with maximum values in the months of May to August. Due to the orographic effect of the Napf range, intense thunderstorms may occur during the summer months. In winter, precipitation may fall as snow, but the development of a continuous snow cover is limited to the higher elevations.





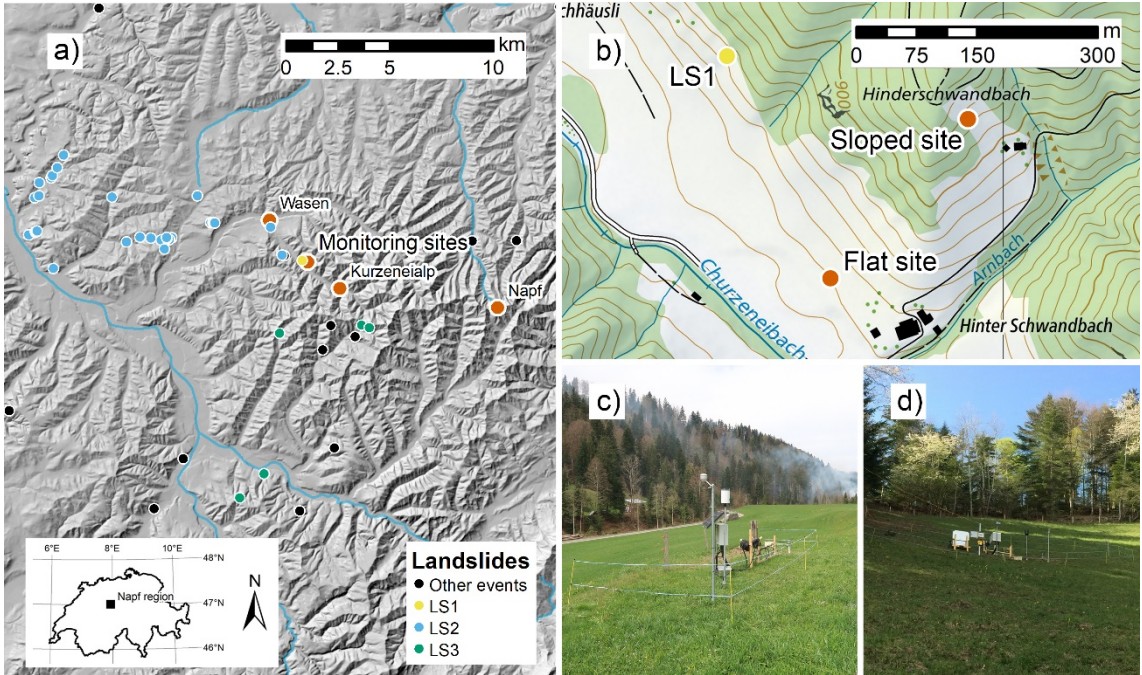

**Figure 1: (a) Hillshade extract of the Napf region showing the locations of the soil wetness monitoring sites ('Monitoring sites'), as well as the meteorological sites Wasen, Kurzeneialp and Napf with orange dots. Further, the locations of landslides that occurred during the study period are shown with yellow, blue, and green dots corresponding to the landslide events LS1, LS2 and LS3 (see section 2.4), respectively, and black dots corresponding to all other landslide events (some landslide events included in this study occurred outside the map extract). (b) Topographic map extract showing the locations of the sloped and the flat site and photos of the flat (c) and sloped site (d). (Copyright by Federal Office of Topography swisstopo, Wabern, Switzerland)**

At the study site, two soil wetness monitoring stations were installed in April 2019 (Table 1). The first station was installed on a landslide-prone hillslope ('sloped site'), which is sloped by 30° and oriented towards SE (Figure 1b, d). The site was chosen as it represents a typical shallow landslide-prone hillslope in the region. The second station was installed at 250 m horizontal distance on an old fluvial terrace ('flat site', Figure 1b, c). Analysis of soil samples taken during the installation of the sensors shows that some lithological differences exist between the two locations (Table 2). On the slope, clay and silt content are generally higher and sand content is lower compared to the flat site. The resulting USDA soil taxonomy classes at the sloped site are sandy loam (0.13 m depth) and loam (0.53 and 0.98 m depth), whereas all soil samples were classified as sandy loam at the flat site. Manual hand auger drillings indicate a depth to bedrock of 1.5 to 2.0 m measured along gravity at the sloped site. At the flat site, the bedrock depth was not reached with the hand auger (3.5 m length), which is due to the situation on an old fluvial terrace. Both sites exhibit grassland vegetation and have similar precipitation amounts and air temperatures on most days in the year (Figure 2). Therefore, the differences between the sites can mostly be attributed to differences in the topographical setting such as slope degree, exposition and shading, and hydrogeology.



**Table 1: Coordinates, topographical, geological and vegetation characteristics of the two monitoring sites.**

| Site | Coordinates | Elevation [m a.s.l.] | Topography | Quaternary | Vegetation |
|------|-------------|----------------------|------------|------------|------------|
| Slope | 47.02486 N, 7.81960 E | 924 | 30° slope | Talus | Grassland |
| Flat | 47.02302 N, 7.81760 E | 829 | Flat terrain | Fluvial terrace | Grassland |

145

**Table 2: Soil properties of the soil samples collected at the two monitoring sites and textural splits according to the USDA taxonomy.**

| Site | Depth [m] | Textural fraction [%] | | | USDA class | Bulk density [g cm⁻³] | Porosity [m³ m⁻³] |
|------|-----------|------------------------|------------------------|--------------------------|------------|----------------------------------|-------------------------------|
| | | Clay (< 2 μm) | Silt (2–50 μm) | Sand (50–2000 μm) | | | |
| Slope | 0.15 | 6.0 | 47.4 | 46.6 | Sandy loam | 1.36 | 0.49 |
| | 0.50 | 7.6 | 48.1 | 44.3 | Loam | 1.49 | 0.44 |
| | 1.00 | 10.9 | 49.4 | 39.7 | Loam | 1.72 | 0.35 |
| Flat | 0.15 | 2.8 | 37.9 | 59.3 | Sandy loam | 1.28 | 0.52 |
| | 0.45 | 3.4 | 29.6 | 67.0 | Sandy loam | 1.49 | 0.44 |
| | 0.70 | 6.3 | 44.6 | 49.1 | Sandy loam | 1.56 | 0.41 |
| | 1.00 | 6.4 | 40.2 | 53.4 | Sandy loam | 1.57 | 0.41 |

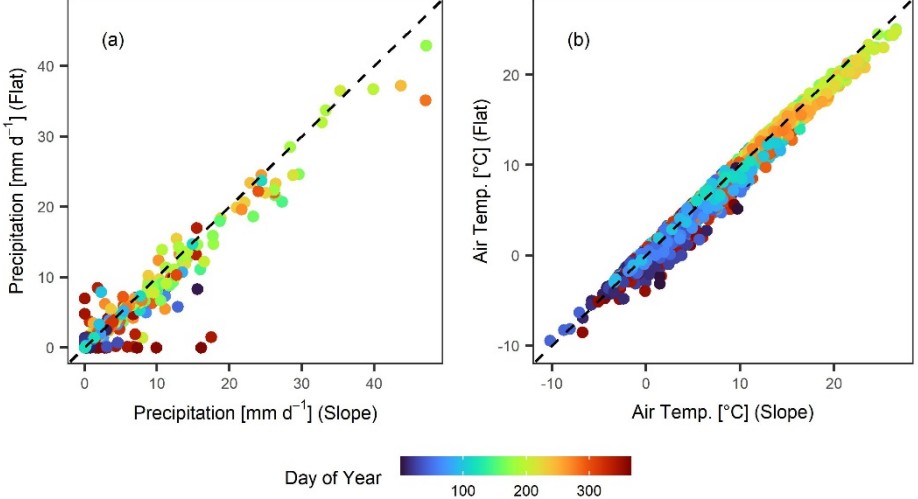

**Figure 2: Sloped vs. flat daily precipitation sums (a) and daily mean air temperature (b). The point colour indicates the day of the**
150 **year. In (a), only days with liquid precipitation are shown (not snowfall) because the rain gauge was heated at the sloped site only.**

## 2.2 In-situ soil wetness measurements

Two different types of in-situ soil wetness sensors were used in this study. Capacitance-based soil moisture sensors (ECH2O
5TE, METER Group) were used to measure VWC. The 5TE creates an electromagnetic field along two prongs by supplying





an oscillating wave at a frequency of 70 MHz. The measurement range of the sensor is 0.0–1.0 $m^3$ $m^{-3}$ with a precision of
±0.03 $m^3$ $m^{-3}$. In this study, we were mainly interested in relative VWC changes. Hence, we did not perform a site-specific
calibration of the sensors. Tensiometers (T8 Tensiometer, METER Group) were used to measure SWP. The T8 measures SWP
with a piezoelectric pressure sensor located in a water-filled porous ceramic cup. The measurement range is -850 to +1000 hPa
and the precision is ±5 hPa. Tensiometers had to be re-filled regularly with degassed water due to the escape of degassed water
after repeated wet-dry-cycles and after reaching suction values below the vaporization point of water, which causes further
degassing of the water.

Sensors were installed at different depths in pairs of soil moisture sensors and tensiometers. At the sloped site, two replications
of each sensor pair were installed at 0.15, 0.30, 0.50 and 1.00 m depth. At the flat site, we attempted the same depth distribution,
however, a gravelly layer between 0.25 and 0.45 m depth prevented the installation of sensors at this depth. Thus, two
replications of each sensor pair were installed at 0.15, 0.50 and 1.00 m depth, and in addition, one sensor pair was installed at
0.20 and 0.70 m depth, respectively. In total, 16 soil moisture sensors and 16 tensiometers were installed at each monitoring
site. Sensors were installed in a soil pit that was dug manually. Following the installation of the sensors, the pit was backfilled
again with the original soil material. During the backfilling process, we considered the original horizonation and layering and
we attempted to reach the original bulk density by manual compaction of the backfilled soil material.

Further to the soil wetness sensors, we installed air temperature sensors and precipitation gauges at each site. At the sloped
site, the precipitation gauge was heated (this was possible due to connection to the electrical grid), allowing for the
quantification of total precipitation amounts even during snow fall. Measurements with all sensors were conducted in 10
minutes resolution and data was transmitted via the mobile phone network every hour. While the flat monitoring site was
powered by a solar panel, the sloped site was connected to the electrical grid. The monitoring stations were installed beginning
of April 2019. In the following analysis, we included data up until end of April 2022, resulting in a total monitoring period of
3 years and 1 month. In general, the monitoring set-up was very robust and only few periods of data gaps exist.

## 2.3 Data processing

First, data quality control was performed in four steps. (1) Outliers were removed by excluding data points outside the
measurement range provided by the manufacturer, and few outliers had to be removed for some sensors by manually defining
a maximum plausible value. (2) SWP values were removed during periods of tensiometer refilling. (3) Periods of unusual
variation and defective periods were removed. This included manual identification and removal of periods with air bubbles
contained in the tensiometers and periods during which the water in the refilling tubes froze. Further, automated correction of
SWP values was conducted at few tensiometers during periods of increased signal noise due to exposition of the refilling tubes
to sun radiation. (4) Long-term data homogeneity was assessed, and periods of inhomogeneity were identified manually and
flagged accordingly.

The second step included the aggregation to hourly values and the normalization of sensor values. Data were normalized for
the calculation of profile mean values, to reduce the effect of lithological differences (in case of VWC) and to reduce the


weighting of dry conditions (in case of the non-linear SWP value distribution in the dry range). We normalized each VWC time series by calculation of the effective saturation (in the following denoted as saturation), S (-), as following:

$$S = \frac{\theta - \theta_r}{\theta_s - \theta_r},$$ (1)

where $\theta$ is the VWC, $\theta_s$ is the saturated VWC and $\theta_r$ is the residual VWC. Here, we used the maximum value of each VWC time series to approximate $\theta_s$, considering that saturated conditions were detected by the tensiometers at each sensor pair at least once during the monitoring period (i.e. positive SWP values were measured). We assumed $\theta_r = 0.08$ m$^3$ m$^{-3}$ from literature values given for the soil types of loam and sandy loam. (Gupta et al., under review in Scientific Data)

In case of SWP, we log transformed dry values (i.e. negative SWP values) due to the non-linear value distribution as following:

$$\psi_{norm} = \begin{cases} -\log(-\psi)) - 1, & \psi < -1 \ hPa \\ \psi, & \psi \geq -1 \ hPa \end{cases}$$ (2)

where $\psi_{norm}$ is the normalized SWP and $\psi$ is the SWP measured by the sensors.

The rain gauges occasionally clogged due to pollen that was collected in the funnel. For the interpretation of the soil wetness data, a continuous precipitation record was produced by filling gaps in the time series of the sloped site with data from the flat site. Here, we chose to gap-fill the sloped site because it was heated and thus provided measurements of total precipitation

amounts. Gaps were filled with measurements from the flat site directly, as the two sites were shown to exhibit very similar precipitation amounts (Figure 2a).

**2.4 Landslide observations**

During the study period, several landslide events occurred in the vicinity of the study site, which permitted to validate the forecast goodness of the landslide forecast model (see section '2.5 Landslide forecast model'). In this study, we gathered

information on landslide events from the Swiss national natural hazard event catalogue 'StorMe' (FOEN, 2022), which includes landslide records from cantonal authorities (based on event documentations) and research institutions (based on review of news articles). Each database entry includes information on the date and time of occurrence as well as the coordinates of the event and most entries additionally include information on the landslide process. Here, we considered all recorded shallow landslide events that occurred within a distance of 40 km of the two monitoring sites. We only included records for

which at least the date of occurrence was known. If the exact timing was not known, it was set to 12.00 h, which was the case for 59 % of the dataset.

During the monitoring period, a total of 217 landslides were recorded within 40 km from the two monitoring sites. Landslides occurred throughout the study period and in both summer and winter (Figure 3b). However, there was a concentration of landslide events during the summer months and particularly during the months of June and July 2021, which was an

exceptionally wet summer in many regions of Switzerland leading to saturated conditions throughout the country (MeteoSwiss, 2021). The precipitation event with the largest number of landslides recorded in the dataset occurred on 24 June 2021. During this event, an intense convective thunderstorm cell triggered a total of 35 landslides within 1.2 to 16.0 km from the study site (LS2 on Figure 1a and Figure 3b). A second notable landslide event occurred on 24 December 2020 within 300 m from the

study site (LS1 on Figure 1a and Figure 3b). In contrast to LS2, only a single landslide event was triggered, and rainfall
amounts were considerably smaller, however it was preceded by intense snow melt in the days before. Finally, in July 2021, a
moderate precipitation event of 17.6 mm over 26 h led to the triggering of four landslide events within 9 km from the sites
(LS3).

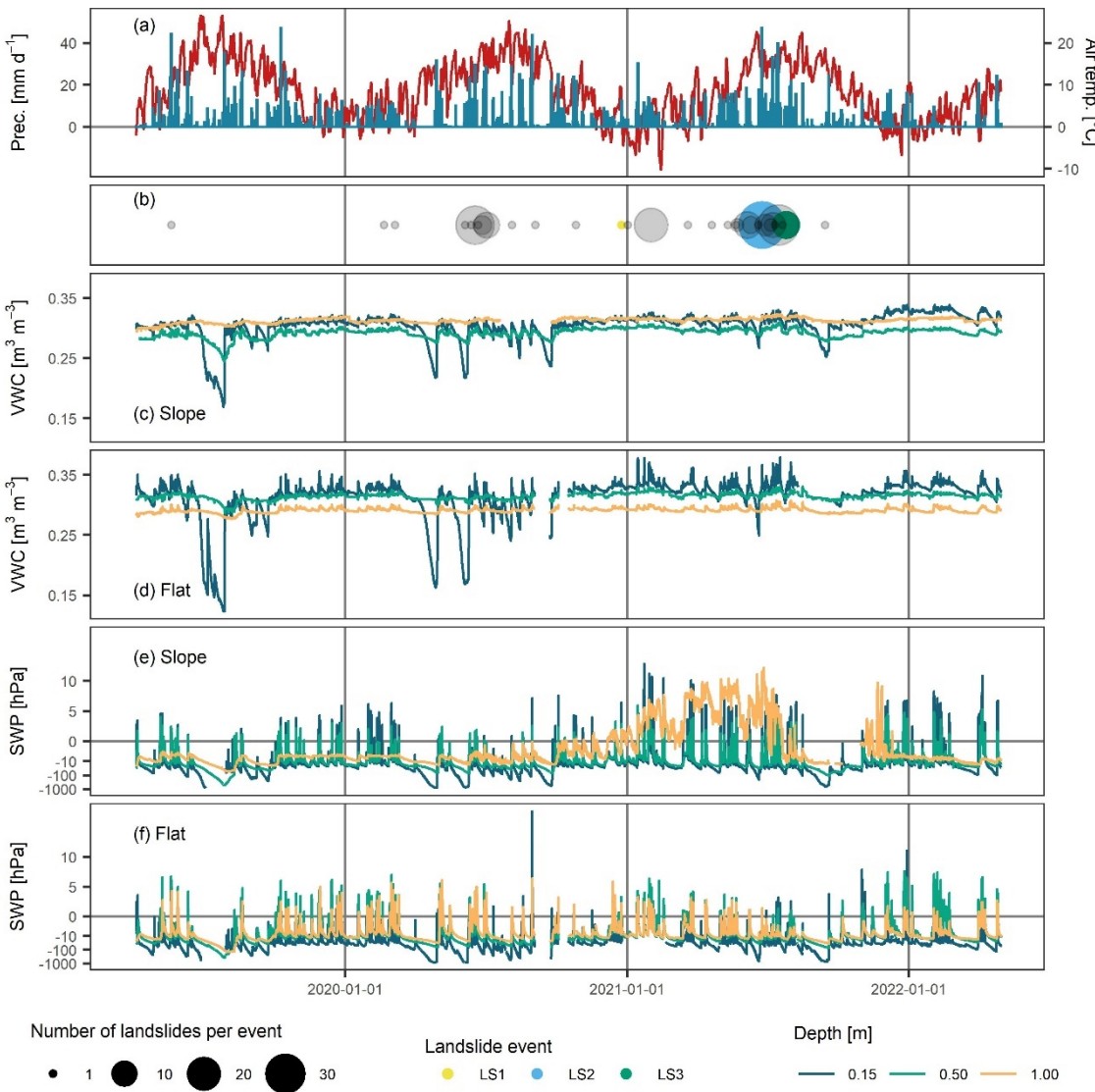

**Figure 3: Time series of daily precipitation (blue) and air temperature variation (red) at the sloped site (a) and number of landslides**
**per event recorded within a distance of 40 km from the monitoring sites shown as filled circles of different sizes (b). The three**
**landslide events LS1, LS2 and LS3 (Figure 1a) are highlighted in yellow, blue, and green colours and all other events are shown in**
**gray. Further, hourly variation of VWC at the sloped (c) and flat site (d) and hourly variation of SWP at the sloped (e) and the flat**
**site (f) at 0.15, 0.50 and 1.00 m depth**





**2.5 Analysis of soil wetness time series related to landslide occurrence**

To directly compare the two sites, soil-profile mean and standard deviation (SD) statistics over multiple depths were calculated for saturation and SWP. To homogenize the different monitoring set-ups with respect to sensor depths, only sensors at common installation depths in both sites were included in the profile statistics (i.e. at 0.15, 0.50 and 1.00 m depth). Further, we included one sensor per installation depth only, which was selected with regards to data quality (i.e. fewest data gaps and best data homogeneity).

In a second step, periods of water infiltration were identified and characterized in the saturation and SWP profile mean time series. To this end, we applied a statistical framework developed by Wicki et al. (2020) and we refer to this publication for a detailed description of the methodology. First, periods of continuous saturation and SWP increase (referred to as 'infiltration events') were identified by an automated algorithm. A unique event identifier was manually attributed to concurrent infiltration events to directly compare events across sites and sensor types. Second, infiltration event properties such as the antecedent 240 wetness (i.e. profile saturation or SWP at the onset of the event) or the wetness increase (i.e. profile saturation or SWP increase during the event), were calculated for each infiltration event. Finally, infiltration events were flagged as landslide triggering or non-triggering depending on whether a landslide occurred at the same time and within a specific distance from the monitoring site (referred to the 'forecast distance'). The landslide triggering classification was conducted for 8 forecast distances (i.e. 5 to 40 km in equal steps of 5 km) to assess the spatial representativeness of the landslide forecast model (see 245 below). Since the exact timing was unknown for many landslide events, infiltration events were flagged as landslide triggering when the respective landslide event occurred within 12 hours prior to and 24 hours after an infiltration event.

To assess the information content of the saturation statistics for the regional occurrence of landslides, an empirical landslide prediction model was applied. The forecast model uses the quantified infiltration event properties as input, and it produces triggering probabilities (i.e. the probability of an infiltration event to belong to the landslide triggering class = 'yes') based on 250 the application of a logistic regression function. The model was previously developed and fitted to a total of 35 soil moisture monitoring stations in Switzerland (Wicki et al., 2020). Note that the two monitoring sites analysed in this study were not part of the model fitting process and the fitting was conducted for a different time period (period 2008–2018). The landslide forecast goodness was assessed by reclassifying the resulting triggering probabilities of each infiltration event to the binary triggering classes 'yes' and 'no' by the application of a threshold, and receiver operating characteristic (ROC) metrics were calculated 255 (according to Fawcett, 2006) by comparison with the observed triggering classification. This process was repeated for 5000 threshold variations to test the overall forecast goodness of a model fit, and it was conducted for all 8 forecast distances. While infiltration events were identified and characterized for both saturation and SWP time series, the landslide forecast model was applied to the saturation-derived infiltration events only, as the fitted model exists only for this type of measurement.





## 3 Results

### 3.1 Temporal soil wetness variation

VWC and SWP values varied at all depths and throughout the study period (Figure 3c–f). In general, VWC and SWP values increased sharply in response to precipitation or snow melt events. They decreased thereafter, initially at a faster rate due gravitational outflow and the progression of the wetting front to the subsurface and later at a slower rate due to evapotranspiration from the surface. Temporal soil wetness variation was highest during the summer months, when high air

temperatures, a fully developed grassland vegetation and sustained periods of no rainfall caused the soil to dry out substantially, intermitted by periods of high intensity precipitation events and strong soil wetness increase. During the winter months, low air temperatures and regular but low intensity precipitation events caused highly saturated conditions at low temporal variation. These seasonal and event-scale characteristic patterns were most pronounced at 0.15 m depth and their amplitudes decreased with depth. At the same depths, the timing and relative magnitudes of variations matched well across sensor types and sites.

Apart from these general fluctuations, differences persisted. Within the SWP time series of the sloped site at 1.00 m depth, a distinct SWP increase was visible from October 2020 until end of July 2021 (Figure 3e), with SWP values becoming positive from January 2021 and reaching a peak in July 2021. This soil wetness increase was not visible at the flat site (Figure 3f). It could be related to an increase in groundwater levels at the hillslope in response to continuously increased water input from fall to summer 2021. However, it may as well be due to a defective sensor. Few data gaps exist, particularly in the SWP time

series during very dry conditions (due to the exceedance of the measurement range) and in winter during cold temperatures (due to the freezing of water in the refilling tubes). Apart from that, data gaps were mostly connected to problems with the electrical power supply of individual sensors and occurred randomly.

    Calculation of the profile mean values allowed for comparison of the two sites in terms of the temporal evolution (Figure 4) or as direct value comparison (Figure 5). Under unsaturated conditions (i.e. $\psi_m < 0$ hPa, S < 1.0), the temporal evolution was

very similar between the two sites and relative event magnitudes matched well. Generally, the flat site appeared drier than the sloped site, however, both datasets approached similar values towards the dry and wet ends of the unsaturated range. We conclude that the sloped site wetted up slower, possibly due to a higher fraction of water flowing off as surface runoff, and that it dried out slower, which we attribute to lower evapotranspiration rates. Under saturated conditions (i.e. $\psi_p \geq 0$ hPa), values scattered significantly (Figure 5b). The values concentrated in two branches indicating that there were periods in which

either the sloped or the flat site experienced highly saturated conditions. As seen previously, high SWP readings were observed at the lowest sensor of the sloped site during the period of October 2020 to July 2021, potentially caused by an increase in groundwater levels or due to a sensor defect. If the respective period is disregarded (to compare periods with the same dominant hydrological processes, i.e., infiltration from above only, Figure 5c), a similar trend is apparent at the saturated end as during unsaturated conditions, with the flat site wetting up faster and to a higher degree than the sloped site. We believe this is due to

a higher fraction of the precipitation or snow melt infiltrating at the flat site compared to the sloped site, where surface runoff is more effective.

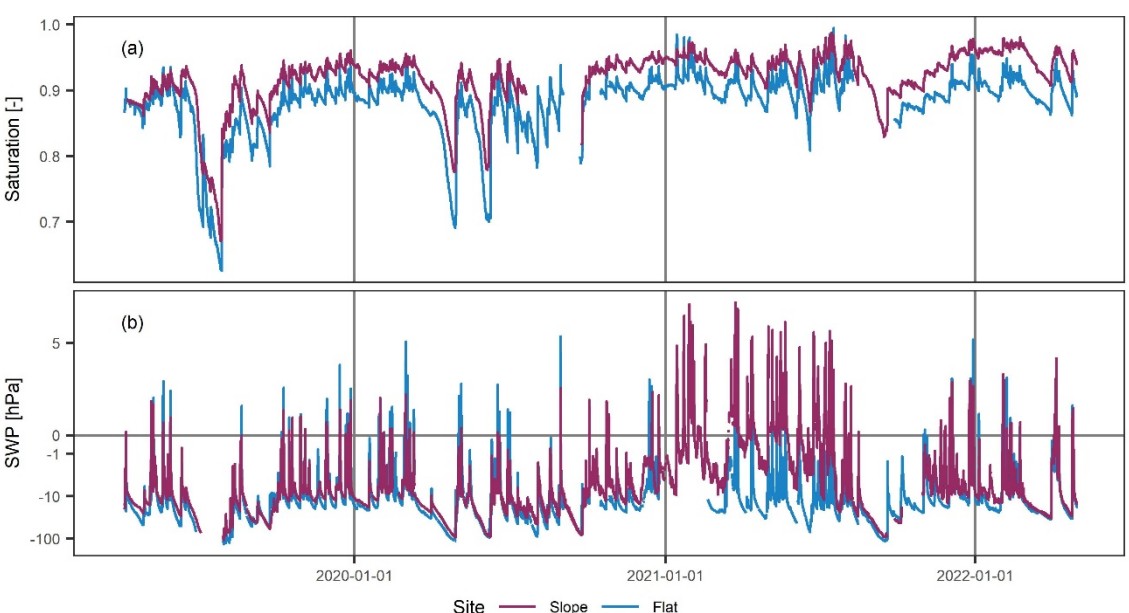

**Figure 4: Hourly profile mean saturation (a) and SWP (b) over the entire study period for the flat and the sloped site.**

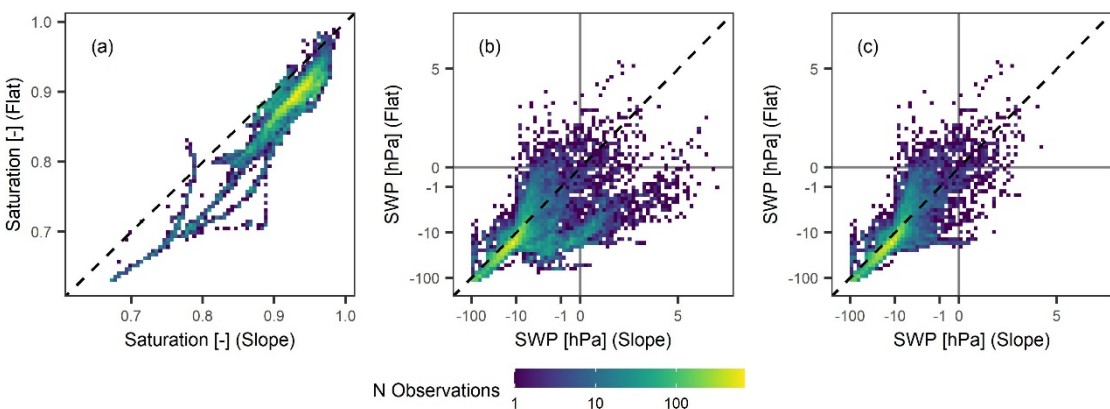

**Figure 5: Hourly sloped vs. flat profile mean saturation (a) and profile mean SWP for the entire monitoring period (b) and disregarding the period 2020-10-01 – 2021-07-20 (c). The colour indicates the number of measurement points (scaled logarithmically).**

## 3.2 Infiltration events

In total, 228 infiltration events were identified in the saturation time series at the sloped site compared to 231 events at the flat
site (Table 3). In the SWP time series, a total of 219 and 203 infiltration events were counted at the sloped and flat site, respectively. Here, fewer events were identified due to the larger data gaps present in the SWP data. Concurrent infiltration events generally occurred at the same time, however onset and end time of infiltration varied particularly after dry periods and during periods of snow melt. The fraction of landslide triggering events (infiltration events with concurrent landslide divided by total number of infiltration events) increased with increasing forecast distance and it ranged from 0.03 (at 5 km forecast





distance) to 0.16 (at 40 km forecast distance) because more landslides were considered for larger distances. In other words, during our observation period every tenth infiltration event caused landslides in a vicinity of 40 km, which is rather exceptional compared to the long-term normal (a landslide triggering fraction of 0.039 was observed for an 11-year observation period including 35 sites in Switzerland; Wicki et al., 2020).

**Table 3: Number of infiltration events identified per site and the fraction of landslide triggering events as function of forecast distance for both saturation and SWP time series, as well as number of landslides observed during the study period per site within a specific forecast distance.**

| Forecast Distance [km] | Saturation | | SWP | | Landslides | |
|---|---|---|---|---|---|---|
| | Slope | Flat | Slope | Flat | Slope | Flat |
| | *Number of infiltration events* | | | | | |
| All distances | 228 | 231 | 219 | 203 | | |
| | *Fraction landslide triggering = 'yes'* | | | | *Number of landslides* | |
| 5 | 0.03 | 0.03 | 0.03 | 0.03 | 16 | 17 |
| 10 | 0.04 | 0.04 | 0.04 | 0.04 | 36 | 36 |
| 15 | 0.05 | 0.05 | 0.05 | 0.04 | 60 | 62 |
| 20 | 0.09 | 0.08 | 0.07 | 0.06 | 90 | 90 |
| 25 | 0.09 | 0.08 | 0.07 | 0.06 | 109 | 110 |
| 30 | 0.10 | 0.10 | 0.10 | 0.08 | 158 | 161 |
| 35 | 0.14 | 0.12 | 0.11 | 0.09 | 188 | 188 |
| 40 | 0.16 | 0.14 | 0.12 | 0.11 | 212 | 217 |

In the following, the few most relevant event properties for landslide triggering as identified in Wicki et al. (2020) are compared between the two sites. These properties either describe the antecedent wetness state of the soil (i.e., antecedent wetness) or

infiltration event dynamics (i.e., wetness change, maximum 3-hour infiltration rate, infiltration event duration). For the saturation dataset, good correlation between the sloped and the flat site was found for the antecedent saturation (correlation coefficient $R^2 = 0.91$, Figure 6a). The flat site showed generally drier antecedent conditions, whereas the differences became smaller towards the wet end. The correlation was less pronounced, and values spread more for the saturation change ($R^2 = 0.59$, Figure 6b). Here, the flat site generally wetted up by a higher degree compared to the sloped site, which we attribute to

more surface ponding and a higher capacity to take up water due to the lower antecedent saturation values. The correlation was even poorer for the maximum 3-hour infiltration rate ($R^2 = 0.28$, Figure 6c), whereas the infiltration rates tended to be higher at the flat site than at the sloped site. Finally, the event duration was similar at both sites inhibiting a good correlation ($R^2 = 0.75$, Figure 6d).

For the SWP dataset, statistical correlations were generally poorer, however they increased significantly if periods of the same

hydrological conditions were considered only (i.e., by excluding the period 2020-10-01 – 2021-07-20, blue points on Figure



6e-h). Correlation was highest for the antecedent SWP ($R^2 = 0.72$ for subset periods, $R^2 = 0.27$ for all dataset, Figure 6e). It was lower for the SWP change ($R^2 = 0.62$ for subset periods, $R^2 = 0.49$ for all dataset, Figure 6f) and for the maximum 3-hour infiltration rate ($R^2 = 0.51$ for subset periods, $R^2 = 0.47$ for all dataset, Figure 6g).

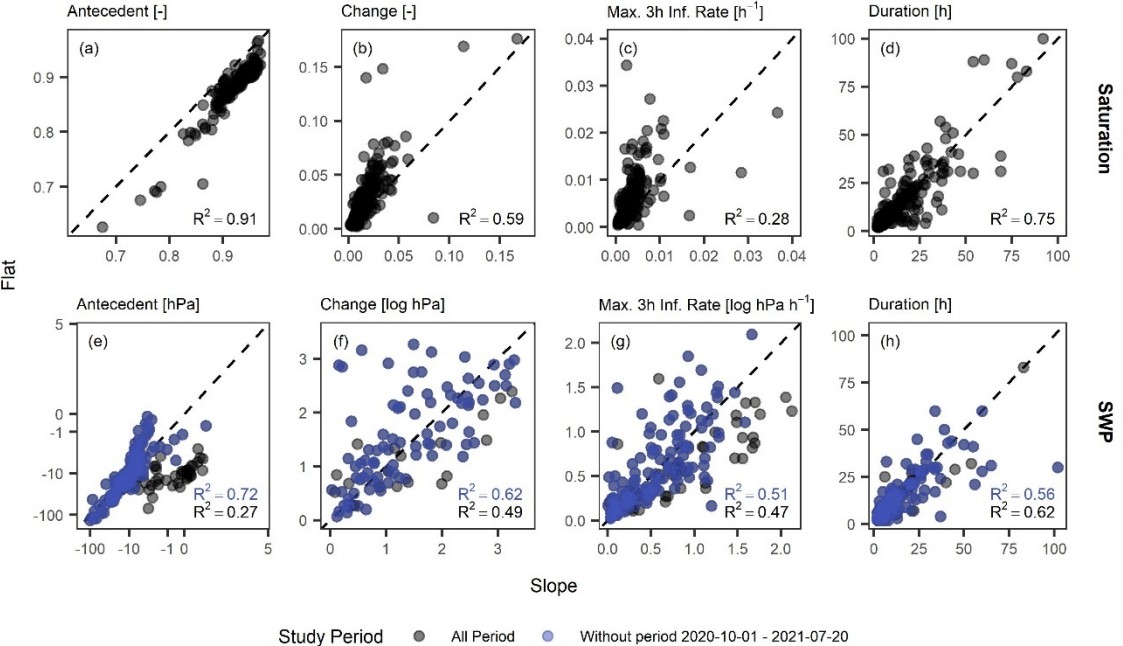

**Figure 6: The infiltration event properties antecedent wetness, wetness increase, maximum 3 hours infiltration rate and event duration for saturation (a-d) and SWP (e-h) showing the slope vs. flat site. The blue points and $R^2$ values in (e-h) disregard the period 2020-10-01 – 2021-07-20.**

Overall, differences in infiltration event properties were in accordance with what was stated in previous sections, i.e. that (i) antecedent wetness conditions appeared drier at the flat site during unsaturated conditions, but that conditions became similar towards saturated conditions, and that (ii) the flat site experienced higher positive porewater pressures during times of full soil saturation if homogeneous periods were considered. Further, soil wetness increase occurred at higher rates and to a higher degree at the flat site, possibly because of more soil water running off on the surface at the sloped site. Generally, differences between the two sites were greater in the saturation time series than in the SWP time series.

### 3.3 Performance of the landslide forecast model

In a next step, the statistical landslide forecast model was applied to the saturation dataset. Here, we applied a model fit that included the two event properties antecedent saturation and saturation change only, as this model showed a high prediction power in Wicki et al. (2020), where it was fitted to 35 soil moisture monitoring sites in Switzerland (not including the two sites of this study). The forecast model has not yet been adapted to the SWP data so far, hence SWP measurements were only used for complementary analysis in this section.



The model output is illustrated for a sample period of 40 days from June 12 to July 22 (year 2021) and for a model fit that is

based on the 15 km forecast distance (Figure 7). At the onset of this period, soil saturation was relatively low due to a sustained

period of no rainfall. However, recurring intense precipitation events over the course of about four weeks led to the attainment

of highly saturated conditions (Figure 7b, d). At the sloped site, saturation levels were generally higher, thus resulting in higher

antecedent wetness conditions at the onset of infiltration, however, saturation increase was lower. Here, this might be due to

the reaching of saturated conditions soon after infiltration started, as indicated by positive pore-water pressures at the SWP

time series.

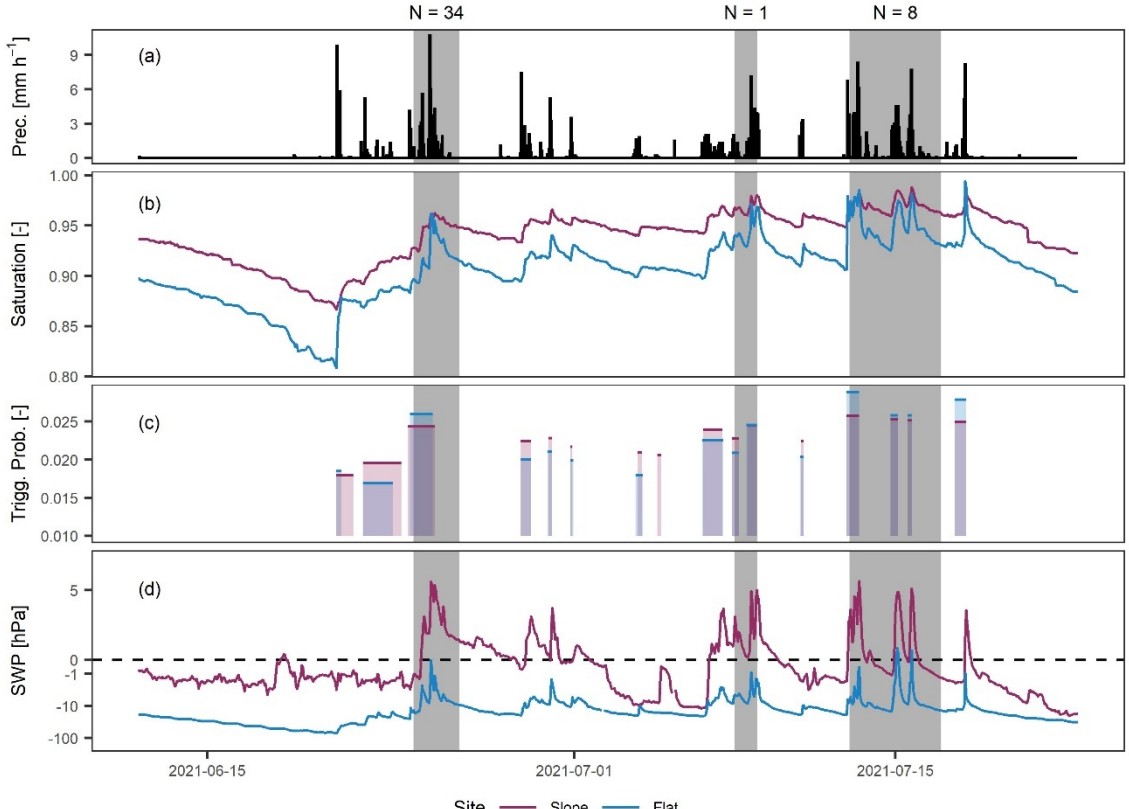

**Figure 7: Time series extract for a period of 40 days in summer 2021 showing hourly precipitation sums at the sloped site (a), as well as the profile mean saturation (b), the triggering probabilities of the statistical landslide prediction model (c), and the profile mean**

**SWP (d) at the two sites. The grey bars indicate consecutive days with observed landslides and the numbers (N) indicate the number of landslide events recorded within a distance of 15 km.**

During the sample period of 40 days shown in Figure 7, three periods occurred when landslides were triggered within 15 km

from the sites (grey bars in Figure 7). All periods occurred during times of highly saturated conditions and they were

accompanied by high positive porewater pressures at the SWP time series. Higher antecedent wetness conditions at the sloped

site resulted in generally higher triggering probabilities (Figure 7c). However, the triggering probabilities at the sloped site

were surpassed at the flat site during few infiltration events due to higher saturation increases. In general, the variation of





triggering probabilities was larger at the flat site due to a higher temporal variation of saturation values. Hence, the triggering
probabilities during the three landslide triggering periods were slightly higher at the flat site too.

Over the entire study period, triggering probabilities scattered less and mean triggering probabilities were higher at the sloped
compared to the flat site (Figure 8). This can be attributed to the generally higher antecedent saturation and lower saturation
change values at the sloped site. Nevertheless, both sites showed a clear distribution difference between the landslide triggering
and non-triggering classes, hence, they were both able to distinguish critically saturated conditions to a certain degree. Yet,
outliers in the form of potential false negatives (too low triggering probabilities in the landslide triggering class) or potential
false positives (too high triggering probabilities in the landslide non-triggering class) are well visible.

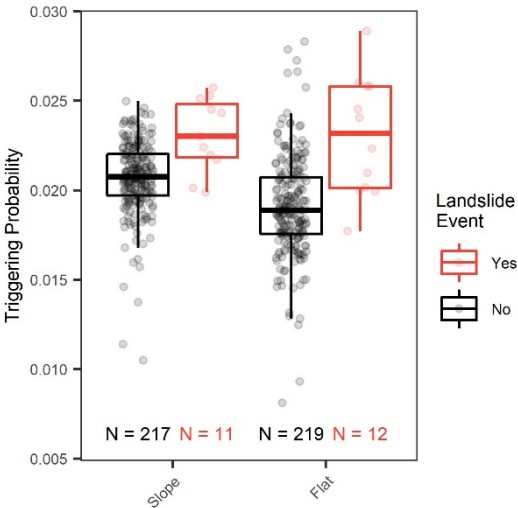


**Figure 8: Boxplots showing the triggering probability distribution of infiltration events without (black) and with landslides (red)
applied to the sloped and flat site separately, for a landslide forecast model based on the antecedent saturation and saturation change
event properties and the 15 km forecast distance. The number of infiltration events is indicated at the bottom.**

Resulting ROC curves clearly deviate from the 0.0 to 1.0 line and most area under the curve (AUC) values are > 0.75 at both
the sloped and flat site and for most of the forecast distances, indicating a good ability to identify critically saturated conditions
(Figure 9a-b). The ROC plots show a decrease in the forecast goodness at short forecast distances (particularly at 5 km), i.e.
in form of large horizontal deviations at the ROC curves. This is due to three landslide events that occurred on July 25 2021
at only 3–4 km distance from the monitoring sites and which were badly predicted by the model. Precipitation radar data for
that day shows that the landslides were triggered by an intense precipitation cell that passed just south of the monitoring sites,
hence the soil wetness signal was not representative for infiltration conditions where the landslides occurred. And since the
number of triggering infiltration events is small at this forecast distance, the resulting model evaluation is less robust (i.e.,
single events have a strong impact on the goodness of fit values). Apart from this, both sites performed similarly, and more
remarkably, both sites showed only a slight forecast goodness decrease with increasing forecast distances. We attribute this to
the fact that during summer 2021 (i.e. when most landslides occurred), saturation was high throughout Switzerland. Thus, the
highly saturated conditions at the monitoring site were also representative for landslide triggering conditions at large distance





from the site. Finally, slight decrease in forecast goodness resulted if both sites were evaluated jointly (Figure 9c), which is due to the different distribution of triggering probabilities at the two sites, however, results appeared more robust.

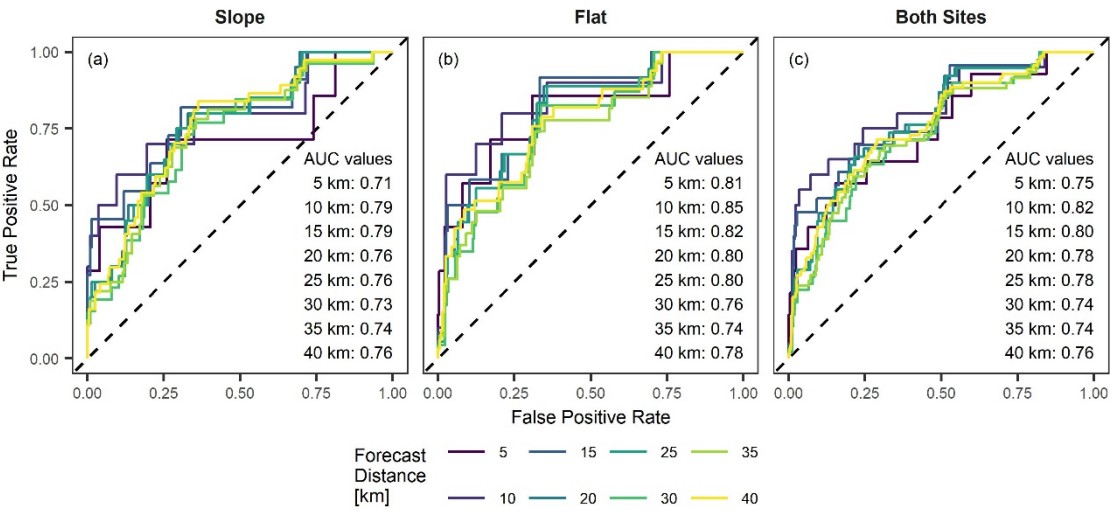

**Figure 9: ROC curves and AUC values at the sloped (a) and flat site (b), as well as at both sites combined (c). Landslide predictions**
**are based on a logistic regression model that predicts landslide probability based on antecedent saturation and saturation change and which was fitted to 35 sites in Switzerland over the time period 2008–2018. The colours correspond to different forecast distances.**

## 4 Discussion

### 4.1 Differences in soil wetness between slope and flat location

Differences in the temporal soil wetness variation between the sloped and the flat monitoring sites can be attributed to different
process domains. The first domain concerns processes that act from the soil surface, such as water infiltration during precipitation and snow melt events as well as evapotranspiration during periods of no rainfall. At the sloped site, conditions were generally wetter, which we partially attributed to lower evapotranspiration rates. These are mainly caused by less total solar radiation input due to shading from neighbouring trees and because the site is more effectively covered from the wind, both factors of which were identified to effectively reduce evapotranspiration rates (e.g. Hu et al., 2009; Li et al., 2005;
McVicar et al., 2007). During rainfall and snow melt events (i.e. during periods of water infiltration) soil wetness increase was less pronounced at the sloped site (i.e. lower increase amounts and rates). This may be primarily caused by the previously discussed higher antecedent saturation levels. Further, decreased infiltration rates at sloped sites were reported to be the result of shallower overland flow depth and thus less submergence of conductive areas (Fox et al., 1997). Finally, the loamy soil texture at the sloped site may cause a lower infiltration capacity (compared to the sandy loam texture at the flat site) based on
soil hydraulic functions. These aspects cannot be validated because site-specific infiltration rates were not measured in the field. The second domain concerns the lateral transport of water in the subsurface, which may be caused at hillslopes by the formation of a transient saturation layer or the rise of a permanent groundwater table above the soil bedrock interface during



stormflow conditions. (Weiler et al., 2005) While the observed SWP increase at the sloped site could be attributed to this, the evidence is weak (measured by one single sensor) and lateral flow cannot be quantified by the sensors used in this study.

The soil hydrological conditions differed considerably between individual infiltration events. In Figure 10 we compare the infiltration process of three landslide triggering infiltration events with depth-varying SWP data. In December 2020, a relatively small precipitation event with a total sum of 12.9 mm within 10 h triggered a shallow landslide event at 300 m distance from the sites (LS1). High pre-event saturation levels throughout the soil profiles (Figure 10a, d) led to fast infiltration of the rainfall water and the build-up of positive porewater pressures at almost all sensor depths. In June 2021, a strong

precipitation event of 61.7 mm over 35 h with peak intensity of 10.7 mm h$^{-1}$ led to the triggering of 35 landslide events within a distance of 16 km from the sites (LS2). While pre-event saturation was considerably lower compared to LS1 (Figure 10b, e), similarly saturated conditions developed until the end of the infiltration event (i.e., positive SWP values throughout the profiles). In July 2021, a moderate precipitation event of 17.6 mm over 26 h led to the triggering of four landslide events within 9 km from the sites (LS3). Pre-event saturation was relatively low particularly at the flat site, thus the infiltrating rainfall only

saturated the uppermost 0.15 m (Figure 10c, f). In contrast to LS1 and LS2, however, indications for saturation from below are visible at the sloped site (i.e., an increase of SWP at the lowest sensor only), possibly because of the activation of stormflow (Figure 10c). During these three landslide triggering infiltration events, soil hydrological differences are larger between individual events than between the sloped and the flat monitoring site. Positive porewater pressures were recorded at most of the landslide triggering events, a condition which was reported in many field-observations of triggered shallow landslides

(Askarinejad et al., 2018; Johnson and Sitar, 1990; e.g. Sidle and Swanston, 1982), however, absolute SWP values measured in this study were still lower than those values reported from other studies where landslides were directly recorded. Further, this comparison shows the importance of characterizing the antecedent state of the soil, as it controls the amount of water needed to saturate a soil column, and as it may effectively lower critical precipitation amounts for the triggering of landslides, which was shown in other studies (Ashland, 2021; Baum and Godt, 2010).




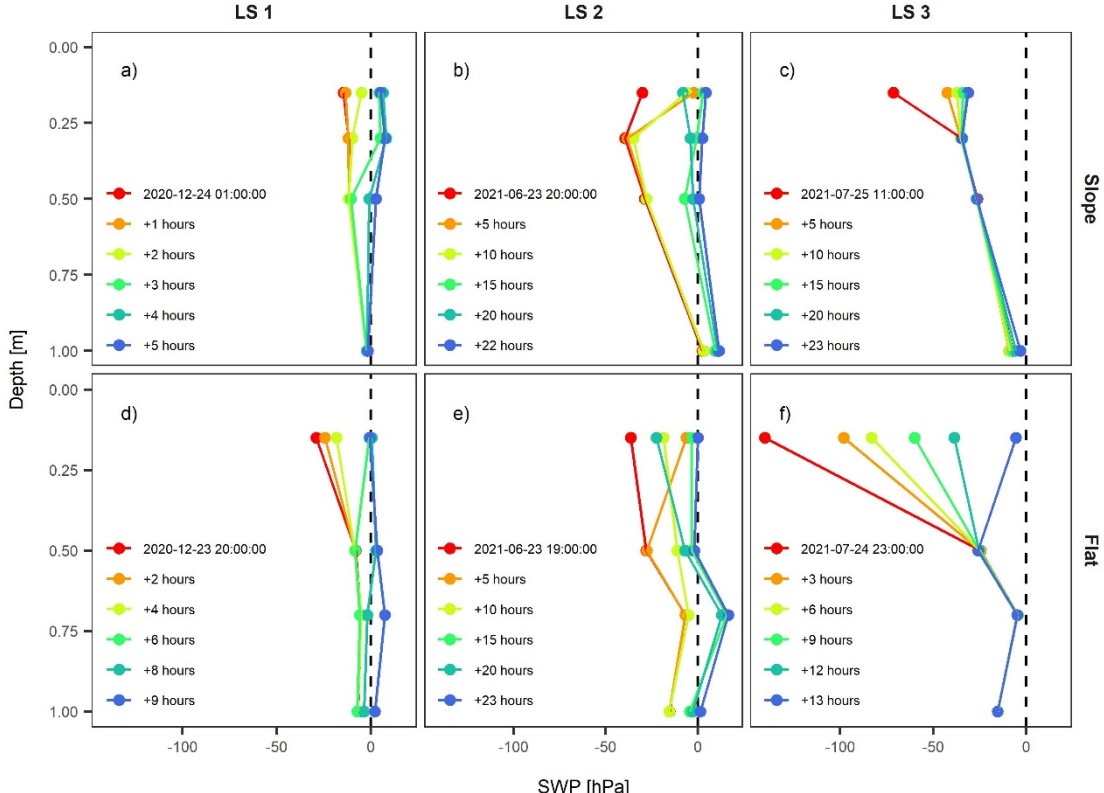

**Figure 10: Depth profiles of soil water potential at the sloped (a-c) and flat site (d-f) during three landslide triggering infiltration events (LS1, LS2, LS3). The first timestep corresponds to the onset of the respective infiltration event. LS1 and LS2 show relatively high antecedent SWP values throughout the profiles and infiltration to large depths, whereas the surface layers at LS3 were relatively dry and infiltration concentrated to the surface layer.**

The above interpretations may be impacted by limitations inherent to soil wetness measurements with in-situ sensors. The measured short-term soil wetness variation by the soil moisture sensors and tensiometers may be impacted by the contact quality of the sensor to the surrounding soil and by the presence of cavities or rocks around the sensor. Both can increase or decrease mean VWC levels, or VWC and SWP variability. (Jackisch et al., 2020) Further, local-scale soil heterogeneities, such as the presence of macropores, may cause preferential infiltration and overprint the resulting signal differences. (e.g. Beven

and Germann, 2013) A methodical constraint is the normalization of VWC and SWP time series, which mainly impacts the calculation of profile mean values. Normalization of VWC time series strongly depends on the chosen maximum (porosity) and minimum values defining water saturation. Here, a fixed minimum value was derived from literature based on lithological criteria. Other studies suggested using the time-series minimum value (Wicki et al., 2020) or normalizing by the porosity value only (Mirus et al., 2018a). We believe a fixed value more reasonable in this case, considering the short observation period and

the similar lithological conditions at the two sites. Finally, in this study, only two sites were measured. For a statistically more





robust result, comparing the data to sites with different topographic configurations (slope, aspect, morphometry) or repeating the experiment in different geological and climatological settings would be valuable.

**4.2 Agreement of soil wetness measurements with prevalent model concepts**

How do our observations conform to prevalent model concepts of soil water distribution in the landscape? To address this
question, we first calculated the TWI for the study area according to Beven and Kirkby (1979). Here, we followed a methodology previously applied to identify spatial soil moisture variation in a mountainous region in the Czech Republic (Kopecký et al., 2021; see Appendix A for a more detailed description of the methodology). The resulting TWI predicts successfully the water courses in the landscape and expected areas of higher saturation (valley floors, concave hillslope areas and gullies, Figure 11). The sloped site has lower values (TWI 3.2–4.0) compared to the flat site (TWI 6.2–6.8), therefore
indicating drier conditions at the sloped site. This classical assumption used in many hydrological studies (e.g. Moore et al., 1991; Sørensen et al., 2006) is by no means reflected in our soil wetness measurements showing wetter conditions at the sloped site. The misrepresentation is mainly caused by the disregarding of the draining conditions (variable depth to bedrock, distance and depth to drainage point) and by consideration of upslope contribution area only.

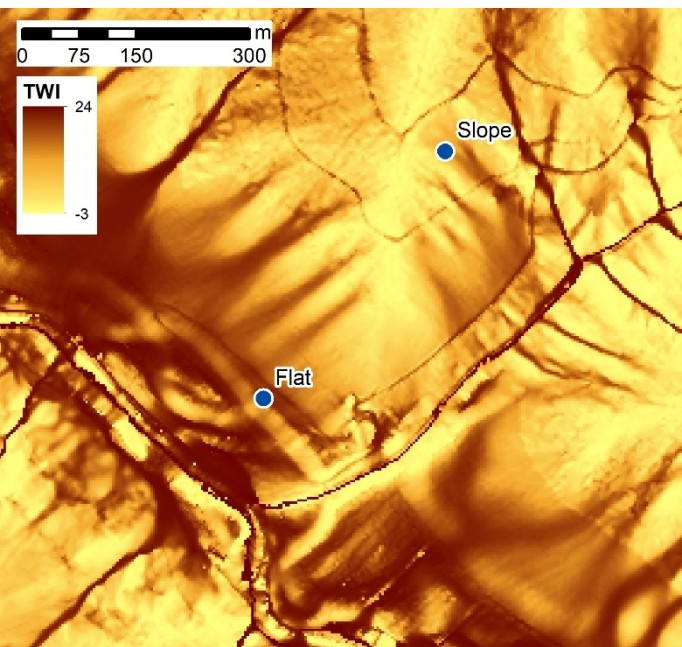

**Figure 11: Topographic wetness index (TWI) at the study sites according to Beven and Kirkby (1979). The extract corresponds to the extent in Figure 1b.**

A better agreement was achieved when comparing our soil wetness observations with a physically-based soil water transfer model. To this end, the temporal saturation variation was simulated using the coupled heat and mass transfer model CoupModel (Jansson and Karlberg, 2011), and the same model set-up, input data and parametrization was used at both sites. We used the
same model set-up as in Wicki et al. (2021) and we refer to this study for a detailed model description. To account for the
different topographic and hydrogeological settings, the lower model boundary conditions were adjusted by explicitly including the elevation difference and horizontal distance to the nearest drainage point. Furthermore, at the sloped site only, an impeding layer was defined at 2 m depth to reflect the shallower depth to bedrock. In addition, the model of the sloped site was coupled to several upslope simulations from which the lateral water output was transferred downslope (resulting in a quasi-2D

simulation of the infiltration process). Finally, the elevation differences between the two sites (~100m) were considered, which mainly impacted the snow cover build-up and melt. Saturation was derived from VWC as described in section 2.3. The simulated temporal profile mean saturation dynamics were similar to the observations from the field monitoring (Figure 12a), with the sloped site showing generally wetter conditions and a lower temporal saturation variation. The coupling of multiple profiles at the sloped site permitted the attainment of lateral water flow (Figure 12b), which was highly variable throughout

the year. Lateral water flow peaked between January and July 2021, which coincides with positive pore water pressure measurements at the sloped site. However, it is evident that the amount of simulated lateral water flow is very sensitive to the number of contributing upslope models (Figure 12b).

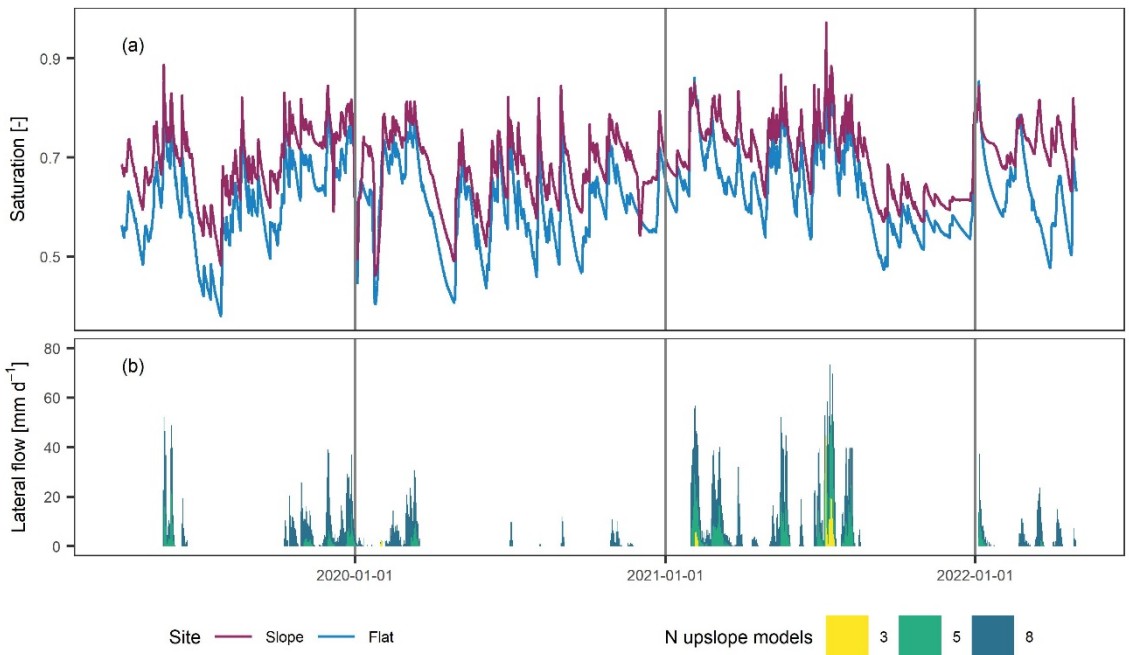

**Figure 12: Simulated soil moisture variation at the sloped and the flat location (number of upslope models defining lateral flow at**
**sloped site was 5) (a) and simulated lateral water flow at the sloped site in the uppermost 2 m as a function of the number of upslope models (b).**

To test the ability of the simulated saturation variation to predict shallow landslides, we applied the same statistical framework to the resulting time series (i.e. identification and quantification of infiltration events and triggering probability prediction). Results show a similar pattern as was observed based on the measurements, with a higher spread of values at the flat site but

similar relative differences between triggering and non-triggering conditions at both sites (Figure 13). These modelling results





demonstrate that the observed differences between the sloped and flat sites (based on the measurements) are systematic and can be well explained by simulations. However, information on hydrogeological settings (draining conditions, depth to bedrock) are important to well reproduce the different topographical settings.

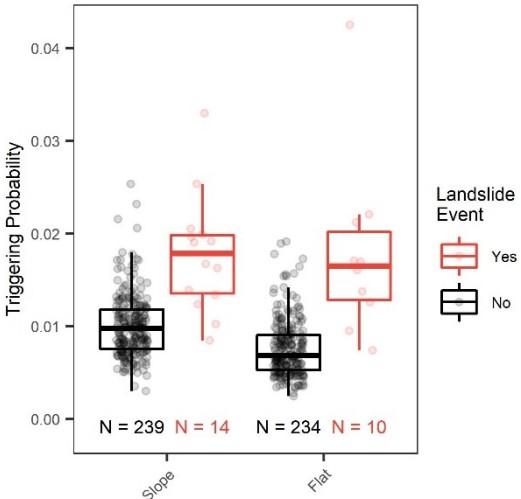

**Figure 13: Boxplots showing the triggering probability distribution of infiltration events without (black) and with landslides (red) applied to the sloped and flat site separately, based on simulated soil moisture (number of upslope models defining lateral flow at sloped site was 5). The number of infiltration events is indicated at the bottom.**

**4.3 Implications for soil wetness monitoring in landslide early warning**

The results of our study showed that measured saturation at both sloped and flat site was well representative for the temporally variable landslide predisposition and triggering conditions in the vicinity of the monitoring site. This encourages the use of (existing) soil wetness monitoring data at flat locations. At the sloped site, additional indication was found for the rise of a shallow or perched groundwater table over the turn of several months. While a rising groundwater table was shown to be an important predisposing and triggering factor for landslides (Van Asch et al., 1999; Iverson et al., 1997), the routing of hillslope water flow is spatially highly variable due to the dependence on bedrock topography and lithological heterogeneities (Brönnimann et al., 2013; e.g. Freer et al., 2002). Installing a soil wetness monitoring site for an operational LEWS at a well-drained location would decrease the impact of such local-scale hydrogeological phenomena and potentially increase the spatial representativeness of in-situ soil wetness measurements for an entire region. The remaining local-scale differences can then mostly be related to topographical characteristics (slope, aspect, shading) and thus potentially be well predicted. Hence, we argue that the local hydrogeological setting may be more important to consider than the topographic setting when deciding for a potential location of a soil wetness monitoring site. Here, we also showed that the overall forecast goodness decreased slightly if both sites were included in the same statistical analysis. This underlines that hydrogeological conditions should be comparable between sites if considered in the same early warning system.





While the causal relationship between pore water pressure and landslide triggering is well known and monitoring of pore water pressures in local LEWSs is well established (e.g. Pecoraro et al., 2019), only few regional landslide forecast models exist to date that use pore water pressure or SWP measurements as direct predictors of regional landslide probability (Pecoraro and Calvello, 2021), or they are mostly used for validation of the hydrological models (Krøgli et al., 2018). Here, we showed that SWP measurements exhibit similar event-scale dynamics as VWC measurements and may add information on perched water tables. This could potentially be used to further discriminate between critical and non-critical conditions for landslide triggering. However, data quality issues due to air bubble formation or due to soil freezing may limit the applicability in an operational monitoring system.

## 5 Conclusions

In this study, we presented time series of soil wetness measurements at a flat and a landslide-prone sloped site over a study period of 3 years. Based on the analysis of temporal soil wetness variation and the comparison with the occurrence of shallow landslides in the region, we conclude following: (i) Differences in the soil hydrological regime between the two sites could be related to hydrogeological and topographical factors. (ii) These differences could be reduced by considering relative changes and by integrating the soil water status over the entire profile, resulting in good correlations between the two sites. (iii) The flat and sloped monitoring site performed equally well in separating landslide triggering from non-triggering conditions. These findings encourage the use of existing soil wetness monitoring sites at flat locations in regional LEWSs. To derive more general conclusions for the planning of future monitoring networks, this analysis should be expanded to additional soil wetness monitoring sites with different hydrogeological settings, explicitly accounting for the local-scale soil hydrological regimes.

## Appendix A

TWI was calculated based on a methodology proposed by Kopecký (2021), which proved useful to identify spatial soil moisture variation in a mountainous region in the Czech Republic. The data analysis was conducted in the open-source SAGA GIS version 8.0.1 (Conrad et al., 2015). As topographical input, we used the swissALTI3D digital elevation model (DEM) in 2 m horizontal resolution (Federal Office of Topography swisstopo, Wabern, Switzerland).
The workflow included following steps:
(1) Pre-processing of the DEM: Filling sinks, setting minimum slope to 0.01°.
(2) Flow accumulation calculation based on the Freeman multiple-flow algorithm (Freeman, 1991).
(3) Calculation of the specific catchment area was with the aspect-based approach by Gruber and Peckham (2009).

**Data and code availability:** The raw soil wetness measurements and the processed profile mean time series will be made publicly available at a repository upon publication.



**Author contribution:** AW installed the monitoring system and conducted the field measurements, analysed the data and
prepared the manuscript. PL, CH and MS supervised the work and reviewed and edited the manuscript.

**Competing interests:** The authors declare that they have no conflict of interest.

**Acknowledgements:** This research project was financially supported by the Swiss National Science Foundation (project
number 175785) and is part of the programme Climate Change Impacts on Alpine Mass Movements of the Swiss Federal
Research Institute WSL. We thank Stefan Boss for the construction of the soil moisture monitoring sites and technical guidance
throughout the project, and we thank Andrea and Lorenz Winkelmann for providing the plot where the measurements were
conducted.

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
