# Peer review of "Impact of topography on in-situ soil wetness measurements for regional landslide early warning – a case study from the Swiss Alpine Foreland"

_Natural Hazards and Earth System Sciences, 2022_

## Author Comment (AC1)

The manuscript deals with an interesting case study about the effect of topography on soil wetness measurements collected in situ for early-warning porpoises. The paper is well structured. The obtained results are sufficiently discussed. Even if dealing with a potentially very interesting topic and its general good writing, in the opinion of this reviewer I recommend minor revision. There are only general comments/questions to the authors:

We thank the reviewer for taking the time to review our manuscript. We thank for the generally positive response and the constructive comments to the individual points raised, which are addressed in the responses below.

1. The introduction part needs to be refined, by adding other specific references in a broader context of the international literature.
   We will provide a broader review of the literature in the reviewed manuscript.

2. The final conclusion to reflect the essential results of the paper is too thin. Please, try to expand this part to give a wide idea of what you learned from the paper, limitations and recommendations of this research should be highlighted.
   We will expand and reformulate the final conclusion part.

3. Authors should discuss also the geographic uncertainty of precipitation data, as both sizes receive roughly the same precipitation amounts (but not exactly the same) even at different elevations.
   Indeed, there are slight precipitation differences, however they are not systematic. Therefore, we don't think there are significant differences between the sites due to elevation or location (the two sites are at only 250 m horizontal and 100 m vertical distance).

4. Regarding the flat monitoring site powered by a solar panel, are there any problems related to low temperatures and exposition?
   We did not experience any problems with powering the flat monitoring site as it receives sufficient radiation throughout the year (the solar panel is south oriented).

5. The data was transmitted via the mobile phone network every hour. My question regards the reception of the data. Did you use a specific internet channel for both sites?
   The station is equipped with a data logger and a modem, which is connected to the GSM network. Every hour, the latest data is gathered on the data logger, and it is sent via the GSM network to the institute where it is copied and stored on an FTP server. We can then access it on that FTP server.

6. Try to add a sketch of the installed sensor instruments and a schematic representation of the monitoring stations.
   We will add a figure with a schematic representation of the monitoring sites to the revised manuscript.

7. Do you think adding different piezometers regarding the rising of the groundwater table for both sites can add the same information in a landslide early warning?
   Rising groundwater tables can be detected by both piezometers and tensiometers as both are capable to measure positive pore water pressures. In comparison to tensiometers however, piezometers may reach larger depths, due to their installation within boreholes. If the borehole is drilled until the soil-bedrock interface, piezometers can be used to detect a temporary groundwater table above the bedrock, which is an indication for hillslope water flow.
   Piezometer measurements were previously shown to be helpful to inform regional landslide early warning systems (e.g. Pecoraro and Calvello, 2021). However, the rise and fall of groundwater levels act at longer time scales than the wetting and drying cycles that are measured with the near-surface sensors. Therefore, data from piezometers and near-surface in-situ soil wetness sensors are not directly comparable. However, it would be interesting to

assess the information content for regional landslide early warning of both measurements if available at the same sites.

In this study, piezometers were originally also installed at both sites. However, at the flat site the soil-bedrock interface was not reached during installation and a ground water level increase was never measured. At the sloped site, the piezometer was damaged early into the campaign due to snow pressure on the borehole tube. Therefore, we were not able to perform such an analysis on the data measured in this case study.

References

Pecoraro, G. and Calvello, M.: Integrating local pore water pressure monitoring in territorial early warning systems for weather-induced landslides, Landslides, 18(4), 1191–1207, doi:10.1007/s10346-020-01599-w, 2021.

---

## Author Comment (AC2)

The manuscript presents a very interesting study on the impact of the topography on in situ soil moisture measurements for regional landslide early warning. The study is well organised and provides a very good overview of the literature. Apart from the analysis and discussion in general, the particularly valuable part of the study is the 3 years of data collected for the specific site and weather conditions, which can be very useful to investigate and understand phenomena other than those considered in the study, and this is something that adds value to the manuscript as well. In general, I think the authors are doing an excellent job in improving our knowledge of rainfall-triggered shallow landslides, and I would like to thank them for their efforts and congratulate them on a very interesting submission.

We thank the reviewer for taking the time to review our manuscript. We thank for the generally positive response and the constructive comments. The specific points raised are addressed in the responses below.

I would have just a few suggestions that I think could be useful in improving the presentation, and also some points that I would like the authors to explain in more detail. Of course, authors are asked to make changes to the manuscript as necessary when addressing these points.

Line 57: In addition to hydrostatic conditions, changes in pore water pressure can also result from the deformations of the slope (interaction pore water - soil structure). Please discuss this point in relation to the pressure potential in general and also in relation to the case study at hand.

We acknowledge that pore water pressure changes can be the result of slope deformations. In this study, we did not find evidence for slope movements or internal deformations at the study sites. Therefore, we believe that this process was not critical for our measurements. However, we will discuss this aspect in the revised manuscript.

Line 71: For certain (coarse-grained) soil textures, residual soil moisture conditions can already be reached at several kPa of matric suction, so this statement may be too general. Please elaborate on this point.

We acknowledge that residual water contents may already be reached at low matric suctions in some coarse-grained textures. This was not the case at this study site. However, we will rephrase the statement in the revised manuscript and mention this aspect.

Line 99: It seems that the word "monitoring" might be missing in the sentence. Please check the sentence and change it if necessary.

We will add the word to the sentence in the reviewed manuscript.

Line 110 (but also relates to lines 142 and 163): This is perhaps the most important limitation in the study and I believe that this point requires appropriate attention - it appears that there are differences in the soils at the two monitored sites and that the differences in soil texture may alter the retention and permeability properties of the soil and affect the measured values to a greater extent than anticipated. Please comment on this point. In this context, please indicate whether hydromechanical characterisation of the soils (e.g. soil-water retention curve, field or laboratory measurements of hydraulic conductivity) has been carried out or whether only the properties given in Table 2 have been determined. It might also be useful to provide a graphical representation of the granulometric composition of the soil with depth for both sites to provide a clearer picture of the materials involved in the study.

We acknowledge that there are differences in soil texture between the two sites. The sloped site is generally finer grained than the flat site (higher clay and slit fraction). However, no other soil

hydrological properties were determined in the laboratory other than the values given in Table 2 of the original manuscript (texture, bulk density, porosity).

To estimate the differences in soil hydraulic properties between the two sites, we have applied a pedotransfer function to the soil samples that were collected at the site. Here we used the Rosetta3 H3w pedotransfer function (Zhang and Schaap, 2017) which predicts soil hydrological properties from soil texture and bulk density values. Owing to the finer soil texture, the sloped site shows slightly lower $K_s$ values throughout the different soil layers (Table A1) and slightly higher values of plant-available water (PAW). Nevertheless, differences are relatively small.

The lower PAW and higher $K_s$ values at the flat site may additionally explain the observed higher rates of drying out. No impact of the soil texture differences on surface runoff can be expected, because the $K_s$ values in the near-surface layers are substantially larger than peak daily precipitation rates (hence, surface runoff is mainly controlled by the antecedent wetness conditions).

We will discuss this in more detail in the revised manuscript and include the soil granulometry in the sketch of the soil profile for better overview.

*Table A1: USDA class and bulk density of the soil samples and soil hydraulic properties derived by the Rosetta H3w pedotransfer function. PTF = pedotransfer function, PAW = Plant-available water.*

| | | Soil Samples | | | | Rosetta H3w PTF (Zhang and Schaap, 2017) | | | |
|---|---|---|---|---|---|---|---|---|---|
| Site | Depth [m] | USDA class | Bulk density [m³ m⁻³] | $\theta_r$ | $\theta_s$ | $K_s$ [cm day⁻¹] | $\theta_{\psi=3.3m}$ [m³ m⁻³] | $\theta_{\psi=150m}$ [m³ m⁻³] | PAW [m³ m⁻³] |
| Slope | 0.15 | Sandy loam | 1.36 | 0.06 | 0.39 | 49.7 | 0.24 | 0.09 | 0.15 |
| | 0.50 | Loam | 1.49 | 0.06 | 0.36 | 27.3 | 0.23 | 0.09 | 0.14 |
| | 1.00 | Loam | 1.72 | 0.06 | 0.32 | 8.0 | 0.22 | 0.10 | 0.12 |
| Flat | 0.15 | Sandy loam | 1.28 | 0.06 | 0.40 | 98.1 | 0.21 | 0.08 | 0.13 |
| | 0.45 | Sandy loam | 1.49 | 0.05 | 0.36 | 64.5 | 0.17 | 0.07 | 0.10 |
| | 0.70 | Sandy loam | 1.56 | 0.06 | 0.35 | 25.1 | 0.21 | 0.08 | 0.13 |
| | 1.00 | Sandy loam | 1.57 | 0.06 | 0.35 | 26.2 | 0.20 | 0.08 | 0.12 |

Line 166: It would be very helpful if you could provide more details about the installation of the sensors (orientation and location/distance from the excavation), a sketch of the installed sensors and also some photos of when they were installed. Please also discuss how the soil disturbance during the installation of the sensors might have affected the measurements.

We will add a figure with a sketch of the installed sensors to the revised manuscript.

Soil disturbance due to the digging and back-filling of the soil profiles may significantly impact the soil moisture measurements. Here, we attempted to reach a similar soil densification during the back-filling of the soil material. We believe, that we reached a sufficient degree of compaction, as no excessive amount of material was left after backfilling and only little settlement of the backfilled material was observed after the installation. Further, we separated the material from the different soil horizons during excavation and backfilled them accordingly again. We will elaborate on this aspect and on the potential impacts on the soil moisture measurements in more detail in the revised manuscript.

Line 169: Please provide the model and manufacturer of the air temperature sensors and precipitation gauges, and include some basic technical specifications, as was the case for the soil moisture and pore water pressure sensors.

We used the following sensors:

- Air temperature sensor: 107 temperature probe (Campbell Scientific), range -35° to +50°C, accuracy ±0.2°C (0° to 50°C range).
- Rain gauge: 52202H tipping bucket (Young), resolution 0.1 mm/tip, accuracy 2% ≤ 25 mm/h, 3% ≤ 50 mm/h, catchment area 200 cm$^2$, rain gauge was heated at the sloped site only, due to power restrictions at the flat site.

We will include this information in the revised manuscript.

Line 178: Please explain in more detail how point (1) of the data quality control was carried out, what the reasons were and also comment on what might have caused such outliers in the collected data (at least in your experience with the sensors used).

Point (1) of the data quality control included two steps:

- Detection and removal of values outside the measurement range which was stated in the technical documentation of the sensors. This was performed for all sensors. Values were removed mostly due to the defect of sensors (very high or very low values over longer periods of time).
- Detection and removal of outliers outside a reasonable range of values (determined visually for individual sensors). This was necessary only for four VWC sensors and very few values were removed. We suspect that a short-term disturbance of the electronic signal may have been the cause, but this is very speculative.

We will include this in the revised manuscript.

Line 182: Please discuss and explain in more detail the problem of solar radiation on the SWP measurements. How does it manifest itself and what was the correction procedure? Please share your valuable experience on how such problems could possibly be eliminated, or include citations or relevant literature if more appropriate, as this issue may be of interest to readers interested in the topic.

Periodically, the tensiometer signal showed an increased amount of noise (up and down of the tensiometer readings). This noise mostly occurred around noon and during the summer months, and it can thus be related to times of increased solar irradiation. Further, the noise occurred irrespective of the installation depth of the sensor. Therefore, we think it is connected to the heating up of a part of the sensor that is located at the surface.

The pressure readings from inside the tensiometer are automatically corrected by the reference atmospheric air pressure, which is conducted to a pressure transducer inside the tensiometer through an air permeable Teflon membrane that is located at the surface. We think that the evaporation of condensed water on this membrane may impact the pressure measurement and thus may be the cause of the observed noise.

We did not find scientific literature where similar problems were reported. However, other users of the same sensor reported similar noise (Viktor Stadelmann, Agroscope, personal communication June 2019). The sensor has since been updated (follow-up sensor: Teros32, MeterGroup), for which the reference air pressure is being measured by a separate air pressure sensor. We have installed the new sensor at other sites, and from our experience, the problem has vanished.

To remove the noise, we automatically identified periods of repeated signal increase and decrease above a threshold with an automated procedure. We then smoothed the signal over these periods (running mean) to remove the noise.

Lines 190 to 193 and line 282: Please explain if the occurrence of surface runoff or water accumulation on the soil surface was observed/monitored in the study (visually or by a sensor) or is this point just generally assumed as a possibility in some scenarios?

Surface runoff is considered plausible at this location, however it was not monitored or observed visually at the field site. We will make this clearer in the revised manuscript.

Line 191: Do the data collected indicate that the soil is affected by hydraulic hysteresis effects? This point seems to have been completely left out of the discussion or literature review - please consider writing a few sentences on this topic as well.

Figure A1 shows the soil water retention curve (daily mean values) for the 2 uppermost sensor pairs at the sloped site (0.15 and 0.30 m) and the flat site (0.15 and 0.20 m), split up by hydrological year and colored by season. From this it is evident, that hydraulic hysteresis does not occur systematically. Some deviations are visible at 0.15 m of the sloped site (fall months in HY19-20; summer months in HY21-22) and at 0.20 m of the flat site (fall months HY19-20). However, because these observations are not systematic (season and depth) we attribute this rather to technical problems with individual soil moisture sensors during specific periods of the study period (the 5TE-sensor partially showed step increases or decreases that were clearly related to technical problems as no precipitation event occurred at the time).

We will mention this in the revised manuscript.

[Figure]

*Figure A1: Soil water retention curve for the 2 uppermost sensor pairs at the sloped site (a–f) and the flat site (g–l) split up by hydrological year (September – August). SON = September, October, November; DJF = December, January, February; MAM = March, April, May; JJA = June, July, August.*

Line 233: It seems that the relative differences in VWC and SWP measurements for the same location and depth (for the same monitored points) are not discussed. Please address this point as well.

Relative differences of sensors at the same depth were largest for the VWC sensors. Here, substantial absolute differences were observed for some sensor pairs. However, the differences in the temporal variability were much smaller. This is a common problem reported for VWC sensors (e.g. Jackisch et al., 2020). To avoid the impact of deviations of individual sensors on the soil wetness signal used for the statistical analysis, we have normalized the VWC values to soil saturation.

Variation in absolute amounts and temporal variability was much smaller between tensiometers at the same depth. Here, normalization was conducted because of the logarithmic scale of the SWP values, which would add weight to dry conditions in the soil wetness signal used for the statistical analysis.

We will discuss this more in the revised manuscript.

Line 266: Drying of the soil during prolonged periods of increased ET without precipitation could also lead to the formation of desiccation cracks in the soil or detachment of the soil from the sensor shaft or loss of hydraulic contact between sensor and soil. Please indicate if desiccation cracks or problems

related to loss of good hydraulic contact between soil and sensor unit were observed during the monitoring period.

We did not observe the formation of desiccation cracks at the two study sites. A dense vegetation cover was present throughout the study period and the regional climate is generally quite wet with the highest average precipitation amounts in the summer. Further, we did not observe indications for loss of hydraulic contact, as a similar temporal variability was observed by both VWC sensors and tensiometers even after dry periods.

Line 274: Given the data redundancy - multiple sensors at a single monitoring point - could these ambiguities be eliminated or at least better understood/explained? Please provide some comments on this point.

Unfortunately, there was only one functional tensiometer at this time at this depth, so there is no redundancy with respect to this measure. Additional measurements include ground temperature (from tensiometers, soil moisture sensors and ground temperature probes), electrical conductivity (from soil moisture sensors) and volumetric water content. However, none of these measurements showed a systematic deviation that could additionally explain the increase in pore water pressure measured by the tensiometers.

Figure 7: It seems that during the dry period (around 20 June 2021) there is a period for the slope when saturation decreases (b) but SWP seems to increase (d) and even reaches positive pwp values. At the same time, precipitation remains absent according to (a). In general, the saturation seems to decrease constantly while the SWP shows a fluctuating behaviour. Please provide an explanation as to what could be the reason for this or an appropriate comment.

In this figure, profile mean values are shown. The reason for generally higher and partially positive SWP values at the sloped site is that the lowest sensor at the sloped site measured constant positive SWP during this period (period of October 2020 to July 2021, as discussed in the original manuscript). At the same time, this sensor showed a larger noise in the signal (as mainly visible during the drying out limbs of the curves). The reason for this increased noise is not known to us and it occurred only during this period. We suspect it might be related to technical problems during measuring positive porewater pressures or it might be related to the technical problems discussed earlier (related to the measurement of reference air pressure).

Line 437: Is it possible that preferential paths for water flow were created when the sensors were installed? Please provide a comment on this point.

Yes, it might be possible that preferential flow paths were created upon installation of the sensors as the soil profile was disturbed upon installation of the sensors (see also comment above). We will discuss this in more detail in the revised manuscript.

Line 536: One of the important values of this study and the research project in general is the data collected. Will data from all VWC and SWP sensors mentioned in the study be available upon publication?

Yes, the data will be provided on a repository upon publication of the manuscript.

Finally, I noticed in several places that the citation follows at the start of the following sentence (e.g. lines 23, 29, 32, etc.). Please check the manuscript and make corrections, if necessary.

We will change this in the revised manuscript.

References

Jackisch, C., Germer, K., Graeff, T., Andrä, I., Schulz, K., Schiedung, M., Haller-Jans, J., Schneider, J., Jaquemotte, J., Helmer, P., Lotz, L., Bauer, A., Hahn, I., Šanda, M., Kumpan, M., Dorner, J., de Rooij, G., Wessel-Bothe, S., Kottmann, L., Schittenhelm, S. and Durner, W.: Soil moisture and matric potential – an open field comparison of sensor systems, Earth Syst. Sci. Data, 12(1), 683–697, doi:10.5194/essd-12-683-2020, 2020.

Pecoraro, G. and Calvello, M.: Integrating local pore water pressure monitoring in territorial early warning systems for weather-induced landslides, Landslides, 18(4), 1191–1207, doi:10.1007/s10346-020-01599-w, 2021.

Zhang, Y. and Schaap, M. G.: Weighted recalibration of the Rosetta pedotransfer model with improved estimates of hydraulic parameter distributions and summary statistics (Rosetta3), J. Hydrol., 547, 39–53, doi:10.1016/j.jhydrol.2017.01.004, 2017.